# Neural Language Models Evaluate Human Performance:
## The Role of Language and Prosody in Predicting Job Interview Scores

## Abstract

In this work we test the use of state-of-the-art neural language model representations to predict behavioral traits that cannot be easily extracted from the textual input alone. We take the task of automated job interview scoring and make predictions on behavioral traits such as *hirability*, *engagement*, or *friendliness*. We find that representing text using neural models trained only on text already leads to better overall prediction results compared to a feature engineering approach that uses a combination of linguistic and extra-linguistic materials. Moreover, we show that combining word embeddings and prosodic features improves the results even further, highlighting the value of adding information from modalities other than text when evaluating human performance.

## 1 Introduction

Recent advances in neural networks have enabled machines to perform tasks such as natural language inference or textual similarity to a very high level of accuracy (Devlin et al., 2019; Wang et al., 2018). A point of investigation relatively new and not completely explored is the ability of neural language models (NLMs) to also capture latent information that is not directly detectable at the lexical level. These are tasks requiring more than the simple inspection of the meaning of words in context because they aim at evaluating humans' performance based both on the communicative form and intent: e.g., the evaluation of communication skills (Rasipuram and Jayagopi, 2016), proficiency levels (Oh et al., 2017), quality of reviews or posts (Danescu-Niculescu-Mizil et al., 2013; Cheong et al., 2019), and engagement level of public speeches (Acharyya et al., 2020). Another clear example is automatic job interview scoring; this task requires the system to identify and analyze linguistic and stylistic information that is critical to successfully evaluate the competence, and consequent hirability, of the candidates (Rasipuram

and Jayagopi, 2018; Nguyen et al., 2014; DeGroot and Gooty, 2009). The evaluation of such systems generally requires paragraph-long responses rather than single words or sentences because it is very difficult to *judge* someone's skills based on responses that are too short (Batrinca et al., 2011).

In this paper, we use job interview scoring as a testbed for our analyses. We investigate the ability of NLMs to process paragraph-long texts and successfully predict behavioral variables such as *hirability* or *friendliness*, by simply using latent information expressed in the language. Our contributions are threefold. First, we use word embeddings generated by state-of-the-art neural language models instead of manually created features as predictors in our task. Second, we show how the combination of word embeddings to capture paragraph level information significantly outperforms existing feature engineering approaches. Third, we perform predictions using regression models instead of classification models, which allows for a more precise comparison between the performance of humans and neural models.

## 2 Background and Related Work

The automatic evaluation of human performance is often carried out by using feature engineering approaches, in which manually extracted features from the lexical, acoustic or visual modalities are selected and fed into the prediction models. Rasipuram and Jayagopi (2016) designed a model that predicts the communication skills of people based on prosody and visual cues from their interview videos. Oh et al. (2017) designed a DNN-based language proficiency assessment classifier that assigns the speech responses of non-native speakers into acceptance or rejection by extracting meaning features and grammar features. Cheong et al. (2019) performed automatic detection of the thoughtfulness of a post in an online discussion forum, using both structural and syntactic features. Zong

et al. (2020) performed a forecasting skill prediction combining textual and cognitive factors and concluded that the textual materials are sufficient for the task. Agrawal et al. (2020), Naim et al. (2018), and Nguyen et al. (2014) performed the task of predicting performance scores of job interviews using several manually crafted features.

## 3 Experiments

### 3.1 Data

We use the job interview dataset provided by Naim et al. (2018), which consists of transcripts, videos, and scores of 138 paragraph-long responses (747 tokens on average) by MIT students in a mock job interview. Below is an example of a potential response taken from Naim et al. (2018).

> "I led the team by showing how to program the robot. The students did a wonderful job! In ten weeks, we made the robot play soccer. It was a lot of fun."

Each response was evaluated by 9 human raters on a scale from 1 to 7 for diverse behavioral traits such as *friendliness*, *engagement*, or *hirability*, many of which require access to information that is not directly available in the linguistic input (see Figure 1 for the full list of traits and Naim et al. (2018) for their description).

### 3.2 Language Modeling and Paragraph Representation

As shown in Table 1, we build the linguistic representations for our experiments by using the output of: a) four static neural language models (word2vec (Mikolov et al., 2014), fastText-wiki, fastText-crawl (Mikolov et al., 2018), gloVe (Pennington et al., 2014)), and b) four different combinations of BERT embeddings (Devlin et al., 2019). We follow Devlin et al. (2019) to select and combine the best four output layers of a BERT-base model.

To represent the entire paragraph, we take the average and the sum of the embeddings of each word in it and produce one vector representation for each paragraph. This approach was inspired by Arora et al. (2017), who reported that this method outperformed more sophisticated approaches for sentence representations by about 10–30%. We expand this approach to paragraph level and create one numerical representation for each paragraph.

### 3.3 Experimental Setup

We treat our task as a regression problem, with the target scores being continuous numbers averaged among the nine human raters. In this work, we perform two experiments and compare the use of information embedded in different neural language model representations alone and in combination with prosodic information against manually crafted features (baseline).

**Baseline** As baseline we take the best performance from Naim et al. (2018), who uses hand-picked features from three modalities (linguistic, prosodic, and facial) as input parameters of two regression models (Lasso Regression and Support Vector Regression). Their linguistic features, 23 of which are obtained from the software LIWC (Tausczik and Pennebaker, 2010), include features like content word categories, positive/negative emotion words, or function word categories. Their prosodic features include information such as fundamental frequency (F0) and intensity. Their facial features include information regarding movements of eyebrows and lips, nods and head-shakes. In all the subsequent comparisons, the baseline results are the ones obtained by combining features from all three modalities.

**Our experiments** We perform two experiments. In *Experiment 1*, only linguistic information –in the form of paragraph embeddings– is used as features for the regression models. In *Experiment 2*, the linguistic information is combined with the prosodic information provided by Naim et al. (2018) to probe for potential improvement in the model performance while providing multimodal information to the system. We exclude facial features from our experiments because, in the original study, the improvement in prediction results obtained by adding facial features was minimal and limited to certain features such as *friendliness* or *excitement* (See Figure 5 in Naim et al. (2018)). This is probably due to the fact that visual features were automatically extracted from the video recordings and, consequently, are extremely noisy. For both experiments, we use Lasso Regression and Support Vector Regression for comparisons of the prediction results against Naim et al. (2018). Due to the small size of the dataset provided, we consider these models the only valid options to obtain reliable results.

**Setup** The prediction experiments are performed using the *sklearn* library in Python. A five-fold cross-validation is performed to avoid overfitting.

| Name | Type | Description and Trained Corpora |
|------|------|-------------------------------|
| word2vec | S | word embeddings produced by word2vec (Google News) |
| gloVe | S | word embeddings produced by gloVe (Wikipedia and Gigaword) |
| wiki | S | word embeddings produced by fastText (Wikipedia) |
| crawl | S | word embeddings produced by fastText (Common Crawl) |
| BERT-all | C | row-wise sum of the weights from all 12 Transformer output layers |
| BERT-s2l | C | the weights from the second-to-last Transformer output layer |
| BERT-4sum | C | row-wise sum of the weights from the last four Transformer output layers |
| BERT-4cat | C | row-wise concatenation of the weights from the last four Transformer output layers |

Table 1: Overview of the 8 word embedding types used in our experiments. The top four embeddings are static embeddings (S) and the bottom four embeddings are contextualized embeddings (C). Finally, each word embedding is either *summed* or *averaged* across paragraphs resulting in a total of 16 representations.

Pearson's correlation coefficient between human-generated and machine-generated scores is used as our evaluation metric as in Naim et al. (2018). The grid search algorithm is used to tune hyperparameters that elicit better results for the majority of the traits.

## 4 Results

### 4.1 Experiment 1

After comparing the performance of the 16 possible paragraph representations as predictors of the two regression methods (Lasso and SVR), we find very similar and consistent results.[1] Because of limited space and for clarity, in the following sections we only report the results from our best combination of models and regression methods: Lasso regression on word2vec (summed) from the static models and BERT-all (summed) from the contextualized models.

As shown in Figure 1, with the exception of *Excited*, *recommendHire*, and *noFillerWords*, our paragraph-based language models outperform the baseline approach with a varying degree per trait. We perform a pairwise t-test to statistically compare the average performance improvement (in the form of correlation coefficients) of our models compared to the baseline. The t-test analysis shows that word2vec (M = 0.71 ± 0.07) significantly outperforms the baseline approach (M = 0.57 ± 0.16, p-value < 0.007). Moreover, as indicated by the big reduction in the standard deviation values, the neu-

ral models obtain more even performances across the predicted individual traits compared to the baseline (SD baseline = 0.16 vs. word2vec = 0.07, BERT-all = 0.10). This indicates that our models are robust on a wider range of traits compared to the feature engineering approach. Compared to the baseline model, which especially struggles for traits like *notStressed* or *eyeContact*, even though the information from the prosodic and facial modalities was leveraged, our static models show significantly better results. Also BERT-all leads to a slight yet non-significant improvement (M = 0.66 ± 0.10, p = 0.07) compared to the baseline although it does not outperform word2vec (p = 0.10). It is worth noting that neural models entirely based on lexical cues significantly outperform a model that combines features extracted from three different modalities. Particularly interesting traits are *notStressed*, *eyeContact*, *calm*, *authentic*, *smiled*, focused, and *paused*, which show highly improved results compared to the baseline even though, intuitively, making judgments on such traits should require more than just textual information.

### 4.2 Experiment 2

In Experiment 2 we test how adding prosodic information affects the overall model performance. We combine linguistic and prosodic inputs by concatenating the corresponding two vector representations. Adding prosodic features to word2vec (word2vec+pro; M = 0.76 ± 0.07) leads to a slight but non-significant improvement (p = 0.10); whereas the addition of prosodic features to BERT-all (BERT-all+pro; M = 0.77 ± 0.08) leads to a significant improvement (p < 0.007). Furthermore, BERT-all+pro does not outperform the language-only representation by word2vec (p = 0.06). This last result indicates that the use of textual represen-

---

[1]The full results for the 8 word embedding types summed and averaged can be found at: http://osf.io/6gzyq/?view_only=700678d4764e4feba545c0dddb0df6f5

[1]To attenuate the problem of multiple comparisons, all p-values have been alpha-corrected: ∗∗ = p-value ≤ 0.007,∗∗∗ = p-value ≤ 0.0007

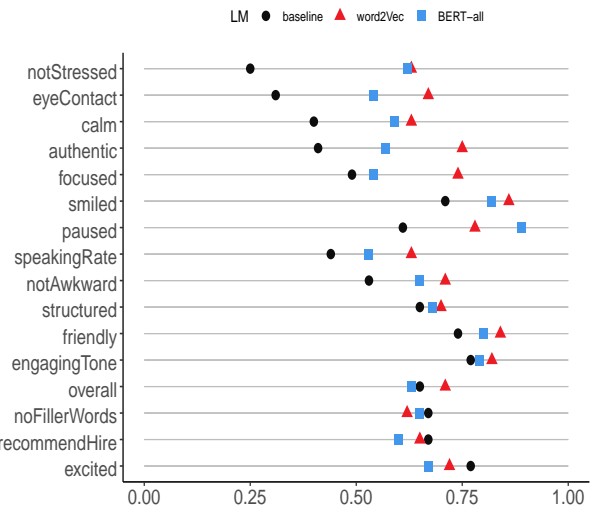

Figure 1: Experiment 1 – Pearson's correlation coefficients for each trait predicted by a Lasso Regression model using: manual features from Naim et al. (2018) (baseline; black circle), word2vec (red triangle), and BERT-all (blue square).

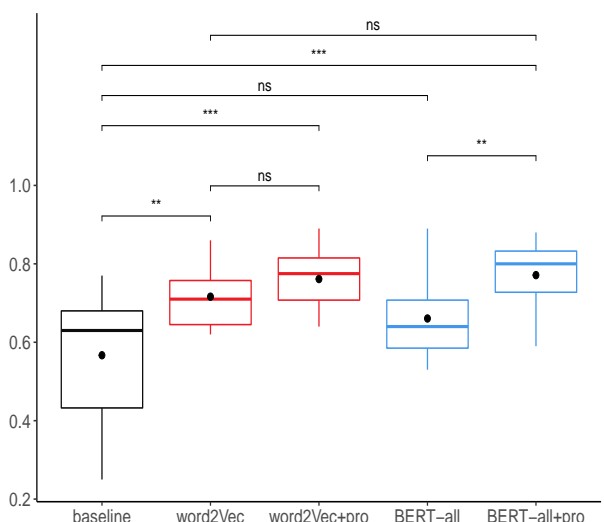

Figure 2: Boxplots representing average Pearson's correlation coefficients across trait for different language modeling types (baseline from Naim et al. (2018) (black), word2vec (red), and BERT-all (blue). word2vec+pro and BERT-all+pro show the average results after combining linguistic and prosodic information. The lines indicate the medians and the dots the mean values across all predicted traits. We provide the significant results from the pairwise t-tests (ns = non-significant, ∗∗ = p ≤ 0.007; ∗ ∗ ∗ = p ≤ 0.0007)

tations obtained by pre-trained word2vec alone is comparable to the use of BERT representations together with the prosodic features (see Figure 2 for statistical significance between different groups). A possible interpretation of these results is that, even though prosody in general plays a contributing role in predicting behavioral traits, its effect becomes more relevant when the linguistic representation alone is not sufficient to perform a specific task and requires external supporting materials.

## 5 Conclusion

Our study shows that neural embeddings generally outperform manually elicited features from multiple modalities in a task that evaluates human performance and on traits that are not easily measurable via shallow access to text. Compared to the feature engineering approach previously adopted, the use of pre-trained embeddings clearly constitutes a step forward in guaranteeing replicability and in reducing implementation issues. Moreover, we show that we can successfully build paragraph-level representations by combining the embedding of each word and still obtain mid-high correlations with human judgments for all the 16 traits (especially with word2vec). Also, our approach performs well on a relatively small dataset, which is valuable given that for many tasks a high amount of data is simply not available or difficult to collect. Finally, we observe that the addition of prosodic features im-

proves the prediction performance even further, especially for models with a lower performance in the language-only setup. This model behavior has an interesting similarity with the way humans process and understand language: when not enough linguistic cues are available at the lexical-semantic level, additional extra-linguistic materials are required to successfully process the information provided (Zhang et al., 2021).

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
