# OpenReview forum: "Neural Language Models Evaluate Human Performance: The Role of Language and Prosody in Predicting Job Interview Scores"
_aclweb.org/ACL/2022/Workshop/CMCL — Submitted to CMCL 2022_

### Official Review · Reviewer_TVRN · 2022-03-22
**Text prediction of hireability**

**Rating:** 4
**Confidence:** 3

**Review:**

This paper describes text-based systems for predicting various qualities related to job interview performance. Systems using word2vec and BERT can outperform traditional feature-based classifiers using small feature sets like LIWC and some basic computer vision applied to interview video. Some scores (including whether the interview candidate smiled) are actually better predicted from text than video. Where differences between w2v and BERT exist, w2v is generally better; the gap can potentially be closed by using prosody.

The work's main strengths are the somewhat surprising fact that some non-textual scores can be predicted from text and the significant improvement over previous work.

I see two main weaknesses: fit with CMCL and misgivings about the paper's impact.

With regard to fit, there is relatively little in this paper that seems cognitive or psychological. The most significant psychological point made is that "when not enough linguistic cues are available at the lexical-semantic level, additional extra-linguistic materials are required to successfully process the information provided", but the argument for this is that prosody improves the weaker BERT classifier more than the strong w2v classifier. Adding extra features generally improves weak classifiers more than strong ones. Moreover, the analysis in the paper does not single out instances where "not enough linguistic cues are provided", but rather ones where the classifier is not making good use of the cues it already has. (As an additional point, failure to find significance should not be considered as an indicator that the effect does not exist, particularly when the models have different variances--- lines 237-244.)

With regard to impact, automated systems for hiring are an ethical minefield (with Amazon's famously sexist hiring tool frequently cited as an example of what not to do). This paper does not consider potential ethical issues resulting from real-world deployment of this technology. Some things that would have to be considered are: the paper makes predictions about qualities such as smiling and speech rate from text which *must be* purely correlational. Such indicators cannot be reported to a decision-maker without explaining that they are imputed without reference to the actual phenomena they claim to be measuring. The paper does not evaluate bias on the basis of demographic characteristics in the classifier output. The current system outperforms the baseline on most of the individual personality factors but not on overall hireability, leading to questions about the validity of this judgment in the first place. While these questions are probably beyond the scope of this work, the paper should include a clear statement that this system cannot be deployed for any practical purpose and a citation to the ethics literature to explain why.

---

### Official Review · Reviewer_9nkW · 2022-03-25
**Predicting hireability from interview transcripts**

**Rating:** 3
**Confidence:** 4

**Review:**

The authors set out to demonstrate that behavioral/personality traits that are deemed relevant for job interview scoring can be predicted on the basis of neural representations of paragraphs of interview transcripts. They compare against a baseline model using hand-crafted features and show that neural representations (static e.g. word2vec and contextual i.e. BERT, also augmented with prosodic information) significantly outperform the baseline. It's somewhat surprising to see this submitted to CMCL, as there is no clear cognitive modeling component to the paper.

It is interesting and unexpected (to this reader) that simple word2vec representations were effectively the best performing representations. I suggest this has more to do with the human raters from whom target scores were obtained (about whom we get no information, e.g. were they trained human resources professionals?) than with the authors' suggestions that prosodic cues are only leveraged when insufficient linguistic cues are available.

I have serious concerns with the ethical implications of this work, especially in light of the fact that ethical considerations are not discussed at all. Technology like this has a very real potential to be misused, but more concerningly often proves discriminatory even when used without nefarious intent. There is a growing literature on the disparate impacts of technology used to assess personal/behavioral traits across demographics. At the very least these concerns need to be addressed in depth in order for a paper like this to be accepted (even setting aside the question of fit for CMCL).

---

### Official Review · Reviewer_WANP · 2022-03-26
**Neural language models predict hireability**

**Rating:** 4
**Confidence:** 3

**Review:**

This paper reports an investigation of how neural language models can be used to judge a candidate's "hireability" and other traits from transcripts of job interviews and writing samples (E1) and prosodic information (E2). They find that embeddings derived from linguistic information alone can outperform the feature-engineered baseline model from previous work which included features for prosody, eye movements, etc. Adding prosodic features results in some performance improvement, but not for the model which already had the highest correlation with human ratings on its own (word2vec).

OVERALL

This is an interesting set of studies, however, I'm not sure that CMCL is the best venue for it. The primary take-away seems to be that neural language models could be used to aid in the process of evaluating candidates, but it's not at all clear what the cognitive/linguistic implications are. If the goal is primarily application, the authors should discuss the important limitations of using these kinds of algorithms to make decisions that impact people's lives, given the well-known biases that they tend to reflect. If the goal is to draw some conclusions about cognition, the link needs to be made much more clear in the framing.


MINOR

The main conclusion suggests that NLMs can be useful for candidate trait evaluation but in fact the evidence presented here only shows that the 4 models highlighted in the main text provide any advantage. It's unclear how many of the model variants actually outperform the baseline. It appears that only a few best-performing ones are statistically compared to the baseline, so the conclusion becomes even more narrow. Again, this would be fine if the goals are about engineering a model that predicts well but unclear what to make of this in terms of cognitive interpretations.

It seems like the choice to only use prosody in addition to embeddings could be better motivated. I am not familiar with the original work (Naim et al., 2018) but it seems like the lack of performance improvement due to facial features in those models doesn't preclude that they could explain meaningful variance in the current models, when paired with a different set of predictors.

---

### Decision · Program_Chairs · 2022-03-29

Reject